# Reversal of Tau-Dependent Cognitive Decay by Blocking Adenosine A1 Receptors: Comparison of Transgenic Mouse Models with Different Levels of Tauopathy

**DOI:** 10.3390/ijms24119260

**Published:** 2023-05-25

**Authors:** Marta Anglada-Huguet, Heike Endepols, Astrid Sydow, Ronja Hilgers, Bernd Neumaier, Alexander Drzezga, Senthilvelrajan Kaniyappan, Eckhard Mandelkow, Eva-Maria Mandelkow

**Affiliations:** 1German Center for Neurodegenerative Diseases (DZNE), Building 99, Venusberg Campus 1, 53127 Bonn, Germanysenthil.kaniyappan@dzne.de (S.K.);; 2Institute of Radiochemistry and Experimental Molecular Imaging, Faculty of Medicine and University Hospital Cologne, University of Cologne, 50937 Cologne, Germany; 3Department of Nuclear Medicine, Faculty of Medicine and University Hospital Cologne, University of Cologne, 50923 Cologne, Germany; 4Forschungszentrum Jülich GmbH, Institute of Neuroscience and Medicine, Nuclear Chemistry (INM-5), Wilhelm-Johnen-Straße, 52428 Jülich, Germany; 5Max Planck Institute for Metabolism Research, 50931 Cologne, Germany; 6Forschungszentrum Jülich GmbH, Institute of Neuroscience and Medicine, Molecular Organization of the Brain (INM-2), Wilhelm-Johnen-Straße, 52428 Jülich, Germany; 7MPI Neurobiology Behavior-caesar, Ludwig-Erhard-Allee 2, 53175 Bonn, Germany; 8Department of Neurodegenerative Diseases and Geriatric Psychiatry, University of Bonn, 53127 Bonn, Germany

**Keywords:** tau protein, Alzheimer’s disease, rolofylline, adenosine A1 receptor, behavior, synapses, PET

## Abstract

The accumulation of tau is a hallmark of several neurodegenerative diseases and is associated with neuronal hypoactivity and presynaptic dysfunction. Oral administration of the adenosine A_1_ receptor antagonist rolofylline (KW-3902) has previously been shown to reverse spatial memory deficits and to normalize the basic synaptic transmission in a mouse line expressing full-length pro-aggregant tau (Tau^ΔK^) at low levels, with late onset of disease. However, the efficacy of treatment remained to be explored for cases of more aggressive tauopathy. Using a combination of behavioral assays, imaging with several PET-tracers, and analysis of brain tissue, we compared the curative reversal of tau pathology by blocking adenosine A1 receptors in three mouse models expressing different types and levels of tau and tau mutants. We show through positron emission tomography using the tracer [^18^F]CPFPX (a selective A1 receptor ligand) that intravenous injection of rolofylline effectively blocks A_1_ receptors in the brain. Moreover, when administered to Tau^ΔK^ mice, rolofylline can reverse tau pathology and synaptic decay. The beneficial effects are also observed in a line with more aggressive tau pathology, expressing the amyloidogenic repeat domain of tau (TauRD^ΔK^) with higher aggregation propensity. Both models develop a progressive tau pathology with missorting, phosphorylation, accumulation of tau, loss of synapses, and cognitive decline. TauRD^ΔK^ causes pronounced neurofibrillary tangle assembly concomitant with neuronal death, whereas Tau^ΔK^ accumulates only to tau pretangles without overt neuronal loss. A third model tested, the rTg4510 line, has a high expression of mutant Tau^P301L^ and hence a very aggressive phenotype starting at ~3 months of age. This line failed to reverse pathology upon rolofylline treatment, consistent with a higher accumulation of tau-specific PET tracers and inflammation. In conclusion, blocking adenosine A1 receptors by rolofylline can reverse pathology if the pathological potential of tau remains below a threshold value that depends on concentration and aggregation propensity.

## 1. Introduction

Alzheimer’s disease (AD) is a neurodegenerative disorder characterized by the progressive accumulation of protein aggregates composed of amyloid-beta (Aβ) and the microtubule-associated tau protein in brain structures, including those relevant for learning and memory [1]. Despite much progress made in recent years with regard to understanding the pathophysiology and the development of novel therapeutic approaches, a number of promising drug candidates failed in clinical trials [2,3,4,5].

The tau protein is well known for stabilizing microtubules in neurons. However, in a subset of neurodegenerative disorders called tauopathies [e.g., Alzheimer’s disease (AD), frontotemporal lobar degeneration (FTLD), Pick’s disease, etc.], tau becomes modified (e.g., by phosphorylation, acetylation, proteolytic processing, etc.), concomitant with neurofibrillary tangle formation [6] which is suspected to induce neurodegeneration. Different aggregation-prone tau mutants have been generated to support this hypothesis [7]. Some examples are the mutation P301L [8,9] and ΔK280 [10,11]. 

Disturbances in the cerebral metabolic rate of glucose (CMRglu) were an early observation in AD [12] and a decrease in glucose consumption is correlated with decreased cognitive performance [13,14,15]. Although reductions in CMRglu in AD patients could be explained at first by the increased synaptic dysfunction and then by the large-scale neuronal loss at later stages, several authors examined the early onset of these defects in people at risk for AD [16]. It appears that hypometabolism occurs in individuals at risk decades before clinical symptoms of dementia are evident, and is not caused by cell loss [17]. On the other hand, such association remains unclear in animal models where inconsistent metabolic activities have been reported, depending on the type of pathology, age, and treatment of the animals [18]. In the case of transgenic mice developed in our laboratory, expressing tau with the aggregation-prone mutation ΔK280 at moderate levels caused hypoactivity of neurons, reduction in neuronal ATP levels and mitochondrial density, loss of dendritic spines, and impaired synaptic functioning [11,19], consistent with decreased glucose uptake.

Adenosine is a neuromodulator and has a depressant effect on neuronal activity when bound to the ubiquitously expressed adenosine A_1_ receptor, a G_i_/G_0_-protein-coupled receptor [20,21]. This makes adenosine a potential therapeutic target for increasing neuronal activity. Indeed, adenosine A_1_ receptor antagonist (rolofylline) treatment restored neuronal activity and rescued previously impaired presynaptic plasticity in vitro in organotypic hippocampal slice cultures expressing mutant tau and in vivo in a tau transgenic model [22]. However, as the Tau^ΔK^ expression in previous studies was low (near physiological levels) and induced tauopathy only at late stages, the question remained whether the treatment would be successful in cases of more aggressive tauopathy. To clarify this, we have now tested two further mouse lines, expressing tau mutants with higher aggregation propensity and at higher levels. The mice were analyzed using behavioral assays, tau-dependent biochemical and histological assays, and PET.

We show here that rolofylline can cross the blood–brain barrier after systemic administration and block the adenosine A_1_ receptor. We also demonstrate that rolofylline-induced recovery of synaptic impairments and cognition observed in the Tau^ΔK^ mice is partially due to a decrease in tau pathology markers and a recovery of synaptic markers (PSD-95) and spines in the hippocampal region. Moreover, we proved that the effect of rolofylline is not mouse line-specific. It produces a beneficial effect in full-length (Tau^ΔK^) as well as repeat-domain (TauRD^ΔK^) transgenic tau-expressing mice. However, there was no recovery of memory deficits or reduction of tau pathology markers in the highly overexpressing and aggressive mouse model (rTg4510). Finally, a second-generation tau tracer ([^18^F]PI-2620) was used for the first time in TauRD^ΔK^ mice, comparing tau pathology with that of the rTgt4510 mouse line.

## 2. Results

### 2.1. Adenosine A1 Receptor Binding of [^18^F]CPFPX Is Reduced by Rolofylline Treatment

Wild-type animals were intravenously injected with 10 mg/kg rolofylline together with [^18^F]CPFPX, a tracer that binds specifically to adenosine A_1_ receptors with high affinity [23]. Mean [^18^F]CPFPX uptake of the whole brain was significantly lower with rolofylline (Figure 1), indicating that rolofylline effectively blocks adenosine A_1_ receptors. Voxel-wise comparison revealed that blocking effects were not homogeneous throughout the brain, but were highest in the midbrain (a region where A1 receptor density is low) and cerebellum (where A1 receptor density is high) [23,24]. VOI analysis demonstrated that rolofylline decreases [^18^F]CPFPX uptake significantly in the midbrain and cerebellum by ~25 and 20%, respectively (*p* < 0.05). A tendency of decreased [^18^F]CPFPX uptake (0.10 > *p* > 0.05) was found in the basal forebrain, frontal cortex, hippocampus, and thalamus (Figure 1E). 

### 2.2. Decreased Tau Accumulation and Synaptopathology after Blockade of Adenosine A_1_ Receptors

Using a mouse model of tauopathy expressing human full-length tau with ΔK280 mutation (Tau^ΔK^), we had shown previously that the adenosine A_1_ receptor antagonist rolofylline (KW-3902) was able to restore spatial memory deficits and normalized the basic synaptic transmission of 14-month-old Tau^ΔK^ transgenic mice [22].

No differences in body weight were observed between groups during the entire treatment (end time point; F(3, 35) = 1.94, *p* = 0.1452). When checking pathology in the Tau^ΔK^ mice after rolofylline treatment, a reduction in pre-aggregated tau species detected by Gallyas silver staining (dash line marks the region of interest corresponding to CA1 cellular layer) and a decrease in conformationally changed tau (Alz-50) were detected in the hippocampus of the mice (Figure 2A). Additionally, levels of human transgenic phosphorylated tau (12E8; epitope pSer262/pSer356) detected by western blot were reduced after rolofylline treatment (Figure 2B for quantification and 2C for representative western blot image). No differences in hippocampal volume were observed in any of the groups in accordance with our previous results [11] (Appendix A). Although no neuronal death was observed in this mouse line, the presence of mutant tau results in a depletion of synaptic proteins in the hippocampus, which was partially reversed after rolofylline treatment (Figure 2D–F). We observed a decrease in the PSD-95 protein levels (Figure 2D) accompanied by a decrease in the number of spines analyzed by Golgi staining (Figure 2E,F) in the Tau^ΔK^ mice, whereas no differences were observed between LCtrls and rolofylline-treated Tau^ΔK^ mice.

### 2.3. Rolofylline Decreases Cognitive Deficits and Tauopathy in Mice Expressing Tau-Repeat Domain (TauRD^ΔK^)

The phenotype of tau pathology in the full-length Tau^ΔK^ described above was relatively mild, correlating with a low expression of exogenous mutant Tau^ΔK^ (~two-fold overexpression of hTau40 over mTau) [11], a moderate propensity for forming β-structured tau amyloid fibrils, and a late onset of cognitive decline (~12 months).

Given this background, we wanted to test the potential of rolofylline in mice with a more aggressive course of tau pathology. We chose the inducible line TauRD^ΔK^, expressing the repeat domain of human tau with the FTDP mutation ΔK280. This domain contains the sequences responsible for the aggregation of tau (PHF1, PHF1* [25]) and has a much higher propensity for β-structure. At an expression level of ~70% or less of endogenous tau [26], the age of decline was earlier (~10 months), the mice had a more severe tangle burden, synapse loss, neuronal loss, and loss of brain volume [27,28,29]. We therefore wanted to investigate whether the beneficial effect of rolofylline was specific to the full-length Tau^ΔK^ mice or could be observed also in the more severe case of TauRD^ΔK^ mice [27].

Fifteen-month-old TauRD^ΔK^ and littermate control mice were treated with rolofylline for 2 weeks at 5 months after cognitive decline started [10]. Cognitive performance was assessed by different behavioral paradigms after rolofylline treatment (Figure 3A). The body weight, reflecting the welfare of the animals, was increased to control levels after blocking adenosine A1 receptor in TauRD^ΔK^ mice (Figure 3B). However, rolofylline did not reverse the ~10% decrease in brain weight observed in the transgenic animals (Figure 3F). In the burrowing test, which is correlated with hippocampal function [30], rolofylline reversed the loss in burrowing behavior back to ~93% of controls (Figure 3C) and was significantly higher than the untreated TauRD^ΔK^ group. Additionally, for fear conditioning testing, the effects of systemic rolofylline administration on different stages of contextual and cue-based (sound) fear learning were investigated 24 h after the training session. Control groups showed a cue-induced freezing response and the rolofylline-treated TauRD^ΔK^ group a clear tendency to increase freezing in the presence of the sound, whereas hardly any increase was seen in the untreated TauRD^ΔK^ mice (Figure 3D). This result shows that the impaired learning association between the sound and the foot shock can be partially reversed by rolofylline treatment. Although the TauRD^ΔK^ mice showed increased contextual freezing during the testing day compared to LCtrl, these mice also presented significantly higher freezing behavior during the training session before the shock (4.73% freezing in LCtrls vs. 24.59% in TauRD^ΔK^ mice; F(3, 35) = 7.189, *p* = 0.0007, Tukey’s post-hoc *p* = 0.0082; Appendix A).

Spatial learning was assessed in the Morris water maze task (MWM), which requires mice to learn to find a hidden platform in a circular pool, guided by spatial cues. During MWM acquisition, TauRD^ΔK^ mice showed a slower rate of learning, which was partially reversed by rolofylline treatment (Figure 3E). No differences were observed in the probe trial of the MWM between control and transgenic animals (F(3, 48) = 0.56; *p* = 0.6412). When checking the distance covered of all groups in the MWM, TauRD^ΔK^ mice also showed a globally slower rate of learning represented by the increased distance covered (F(3, 195) = 4.87, *p* = 0.0027, Tukey’s post-hoc LCtrl vs. TauRD^ΔK^ mice *p* = 0.0214; Appendix A), although no statistical differences were detected when checking specific time points. As a control, mice were tested for their general motor and exploration behavior in the open field test. There was no significant difference between groups (Appendix A), so we can exclude that the difference in learning rate was attributable to a difference in neuromotor disability.

Finally, rolofylline treatment improved the expression of the postsynaptic protein PSD-95 in TauRD^ΔK^ mice (Figure 3G,H), similar to what we observed in the Tau^ΔK^ mice treated with rolofylline, and similar to the reversal after switching off the expression of the transgene in regulatable mice [11]. When analyzing tau pathological markers, rolofylline treatment promoted a reduction in the number of Gallyas silver-positive NFTs in the cortex of TauRD^ΔK^-treated mice as well as a decrease in human transgenic phosphorylated tau (12E8; epitope pSer262/pSer356) checked by immunostaining (Figure 4).

All of these results indicate that even though cognitive abilities are lost in the tau-transgenic TauRD^ΔK^ mice, the loss can be reversed by rolofylline treatment. This holds not only for the relatively mild phenotype of the full-length Tau^ΔK^ expression, but also for the expression of the more aggressive repeat domain TauRD^ΔK^. We note that, in both cases, the pathology can be reversed by “genetic” treatment (i.e., by stopping the inducible expression of mutant tau by addition of doxycyclin); however, none of the tau-directed drugs tested so far (e.g., aggregation inhibitors [31,32] have achieved a reversal, with the exception of rolofylline [22].

### 2.4. Rolofylline Does Not Improve Behavior in rTg4510 Mice Strongly Overexpressing Mutant Tau^P301L^

In the mouse lines described above, tau pathology sets in at late time points, roughly corresponding to stages of AD. There are, however, other tau transgenic lines with stronger phenotypes designed to allow a faster analysis of tau pathology. This includes lines 3xTg-AD by LaFerla and colleagues, which combine several AD-related mutations [33,34], and the line rTg4510, which expresses human 0N4R-Tau with the P301L mutation at a high level (~13× endogenous tau) and in an inducible fashion, created by Ashe and coworkers [8,9]. This line has become a popular model, as it phenocopies the tau pathology and the pronounced neurodegeneration of human tauopathies with an early onset (~3 months; however, note the recent re-analysis showing that this mouse contains several other genetic changes [35] (see Section 3).

To select the time window for a therapeutic rolofylline treatment in this transgenic mouse model, we studied histopathological features during the expression of Tau^P301L^ at different ages. We observed high expression of Tau^P301L^ with development of sarkosyl-insoluble tau already at 3 months of age (Appendix A), whereas Gallyas silver staining of neurons appeared later (Appendix A).

In AD and other tauopathies, tau appears to undergo phosphorylation and conformational changes, accompanied by the translocation of pathological tau from the axonal to the somatodendritic compartment of neurons [36,37]. These changes are also observed in rTg4510 mice as early as 3 months of age, viz. hyperphosphorylation [AT8 antibody, epitope pS202/pT205) and PHF-1 antibody (epitope pS396/pS404)] and pathological conformation of tau (MC1 antibody) appearing in the hippocampus and the cortex (Appendix A). The number of cortical neurons bearing highly phosphorylated tau (e.g., AT8- and PHF1-Tau) and tau conformational changes (MC1) increase progressively with age (Appendix A). At 6 months of P301L expression, mice show histopathological features of AD, including neuronal loss in the CA1 layer of the hippocampus (by NeuN staining, Appendix A). Finally, Iba-1 and GFAP staining against activated microglia and astrocytes, respectively, indicate inflammatory processes in the hippocampus of rTg4510 mice already at 3 months of age (Appendix A).

Rolofylline treatment was tested on these mice for its beneficial effect using similar procedures as those for the mouse lines described above. Based on the histopathological and biochemical results (Appendix A), we selected 5 months of age for the therapeutic treatment because at this age the rTg4510 mice present extensive tau pathology and also cognitive deficits. In this therapeutic treatment protocol, rolofylline application started at 5 months of age for 2 weeks in rTg4510 mice and control littermates, and cognitive assessment with continuous rolofylline treatment was performed using several behavioral tests (Figure 5A). First, we analyzed the body weight without treatment as an assessment of the general wellbeing of the animals. The rTg4510 transgenic mice had reduced body weight compared to their littermates (−12%) as early as 2.5 months of age, although it didn’t gain statistically significance until 4.5 months of age (Figure 5B). There was no improvement through rolofylline treatment (Figure 5C). In fact, surprisingly, rolofylline increased body weight in control animals, an effect not seen with the rTg4510 tested. The rTg4510 mice also showed reduced brain weight, which could not be reversed by rolofylline (Figure 5D). In the nesting and burrowing tests, rTg4510 mice performed significantly worse than control animals with no reversal after blocking A_1_ receptors (Figure 5E,F). Indeed, rolofylline seemed to inhibit nesting behavior in these transgenic animals (Figure 5E). In the open field test, no differences were observed between non-treated control and transgenic animals, whereas rolofylline induced pronounced hyperactivity in rTg4510 (Figure 5G). In the Y-maze test, control mice spent more time in the novel arm, in contrast to rTg4510 mice, which did not show any arm preference (Figure 5H), showing no beneficial effect of rolofylline in restoring spatial memory in these animals.

Consistent with the effects observed in behavior, rolofylline treatment in rTg4510 mice at 5 months of age could not prevent or improve molecular markers of tau pathology. The rTg4510-treated mice showed a similar degree of tau hyperphosphorylation (PHF-1 (pS396/pS404); Figure 6A), tau pathological conformation (MC1; Figure 6B), and tau aggregation (Gallyas silver positive cells; Figure 6C).

Since no recovery was observed in rTg4510 mice after therapeutic treatment, we tested the potential of rolofylline to prevent cognitive decline. For the preventive intervention, rolofylline was initiated at 2.5 months of age for 2 weeks and continued throughout the behavioral experiments (Figure 7A). The rolofylline treatment had no effect on behavioral assays, similar to what we observed in case of brain weight, which was still reduced in rTg4510 mice compared to control mice after treatment (Figure 7B).

When analyzing cognitive decline, the rTg4510 mice performed significantly worse in the nesting test when compared to control littermates regardless the rolofylline treatment (Figure 7C). Just at 12 h after the introduction of the nestlet, rTg4510-treated mice were performing not significantly differently than controls, although there was a clear trend to show reduced nesting behavior (*p* = 0.0569). Likewise, no improvements were observed in the MWM. Transgenic animals presented higher latency to escape in the learning phase, showing that they could not remember where the hidden platform was located. The same results were observed in the rolofylline-treated transgenic animals (Figure 7D). When checking the long-term memory 72 h after the training phase, no differences were observed between treated and non-treated transgenic animals (Figure 7E). Finally, in the open field test, contrary to the assays described so far, no differences were present between groups when analyzing locomotor activity (Appendix A), even though rolofylline-treated rTg4510 animals showed increased anxiety, as shown by the lower exploration of the center of the arena (Figure 7F).

In summary, in the case of rTg4510 mice, neither therapeutic nor preventive rolofylline treatment could reduce the cognitive decline or prevent or reverse markers of tauopathy, in pronounced contrast to what we observed for the Tau^ΔK^ and TauRD^ΔK^ lines.

## 3. Discussion

### 3.1. Reversal of Tau Pathology in Mouse Models with Near-Physiological Expression of Tau Mutants

In the present paper, we assessed the potential beneficial effect of the adenosine A_1_ receptor antagonist rolofylline in transgenic mouse models of tauopathy. For this study, we used three different transgenic mouse lines: a transgenic mice expressing full-length pro-aggregant human tau (2N4R Tau-ΔK280, termed Tau^ΔK^) [38] or its repeat domain (TauRD^ΔK^) [26] and the rTg(tauP301L)4510 (rTg4510) [8,9]. We showed that Tau^ΔK^ and TauRD^ΔK^ mice with moderate levels of aggregation-prone tau protein present hypoactivity of neurons with a reduction in neuronal ATP levels and mitochondrial density, loss of dendritic spines, and impaired synaptic functioning [11,19,22,39]. Rolofylline, a drug developed for patients with acute heart failure and renal dysfunction, normalizes presynaptic functioning and neuronal activity, as well as reducing the number of spines in the in vitro model of Tau^ΔK^ organotypic hippocampal slices. Moreover, when administrated orally to Tau^ΔK^ and TauRD^ΔK^ transgenic mice, rolofylline restores cognitive functioning, which is known to be subdued as a result of adenosine signaling [22] (Figure 3). On the contrary, rTg4510 mice do not improve after rolofylline treatment (Figure 5, Figure 6 and Figure 7).

Adenosine, when bound to the A_1_ receptor, has an inhibitory function on many organs including the brain where it reduces neurotransmitter release [20,21]. These effects can be explained by the coupling of A1 receptor to G_i/o_ which downregulates adenylyl cyclase, thus reducing cAMP and cellular metabolism. Postsynaptic activation of adenosine A1 receptors causes decreased cellular excitability [40]. In the hippocampus, adenosine is predominantly inhibitory due to high expression levels of the high-affinity A_1_R [41], and in vivo evidence showed that prolonged A_1_R activation may lead to increased hippocampal neuronal death [42]. Moreover, a continued intravenous infusion of a high dose of rolofylline for the treatment of heart failure was shown, in some cases, to induce epilepsy [43], suggesting that a systemic administration of rolofylline can cross the blood–brain barrier (BBB). Using a selective A_1_ receptor PET ligand [^18^F]CPFPX, we showed for the first time that rolofylline can cross the BBB and successfully engage the target (adenosine receptors) when administered systemically, since intravenously injected rolofylline was able to reduce the binding of [^18^F]CPFPX to adenosine A_1_ receptors in the brain.

Using the mouse model of tauopathy Tau^ΔK^, we have previously shown the beneficial effect of the adenosine A_1_ receptor antagonist rolofylline in Tau^ΔK^ mice and in organotypic hippocampal slice cultures [22]. In these mice, the tau protein shows the typical hallmarks of a pathological state—it is missorted into the somatodendritic compartment, phosphorylated at key sites (e.g., AT8-epitope), and folded into a pathological conformation (MC-1 epitope) [38]. Mice present cognitive impairment from ~12 months onward despite the near-absence of mature neurofibrillary tangles [11]. Rolofylline treatment clearly reduced the Gallyas-silver positive cells in the hippocampus of old mice with Tau^ΔK^ and TauRD^ΔK^ expression, reduced the level of misfolded tau (Alz50 staining) as well as the levels of Alzheimer-like phosphorylation (12E8; epitope pSer262/pSer356), both indicators of subsequent formation of insoluble NFTs [44].

Remarkably, the reversal of cognitive decline upon rolofylline treatment was observed not only in full-length Tau^ΔK^ mice [22], but also in TauRD^ΔK^ mice expressing only the mutant repeat domain, which has a much higher aggregation propensity and earlier onset of cognitive decline. The overall recovery of body weight in this mouse line was also a marker for improvement of general welfare. Tau-related pathological changes predominantly affect the hippocampal formation in these mice [29]. The hippocampus plays an important role in spatial and contextual learning processes and memory consolidation after initial acquisition [45]. To examine hippocampus-dependent learning and memory, the MWM test, burrowing test, and cue and context fear conditioning paradigm were used [30,46,47]. Different types of hippocampal lesions can decrease burrowing rate [30], which was shown to be altered in TauRD^ΔK^ mice that also exhibited longer latency to reach the hidden platform in the MWM. These acquisition deficits were not due to impairment in neuromotor performance, as no impairment in the open field test was observed. In both tests, rolofylline treatment reversed the loss of cognitive performance. With a more subtle effect, the same was true for cue-based (sound) fear learning, where rolofylline-treated transgenic animals showed a clear tendency to increase freezing behavior in the presence of the cue. All these reversal effects, achieved by an initial 2 weeks of rolofylline treatment (and continuous treatment during behavior assessment), were analogous to those previously reported for the Tau^ΔK^ mouse line [22], in spite of the higher pathogenicity of TauRD^ΔK^, underscoring the curative potential of the adenosine A_1_ receptor antagonist.

The causal link between adenosine (an extracellular ligand) and mutant tau (a cytosolic protein) is currently unknown. One of the various effects of mutant tau is the reduction in cellular energy in the form of ATP, suggesting a link to mitochondrial function [22]. In neurons, tau is particularly located to presynapses where it is linked to synaptic vesicles via synaptogyrin3 [48]. In this context, it is notable that incipient synaptic degeneration leads to activation and migration of microglia, releasing ATP that is converted to adenosine and hence downregulates cAMP via A1 receptors [49,50]. In this scenario, rolofylline could counteract adenosine and thus prevent synaptic exhaustion, as observed.

### 3.2. No Reversal of Tau Pathology in Mouse Models with High Non-Physiological Expression of Tau Mutants

In contrast to the mouse lines above with near physiological levels of mutant tau, a different picture emerged for the rTg4510 mice, which express much higher levels of aggregation-prone tau (~13× endogenous tau) [8,9] than the Tau^ΔK^ and TauRD^ΔK^ mice (~2× and 0.7× endogenous tau, respectively) [11,26]. This line displays pronounced tau pathology and cognitive decline at early stages (~3 months) [8] (Appendix A and Figure 7) and therefore reaches the irreversible “point of no return” much earlier than the Tau^ΔK^ and TauRD^ΔK^ lines where tau protein aggregates more slowly and at lower expression levels. Indeed, neither a therapeutic treatment with rolofylline (of 5-month-old mice, after onset of cognitive decline) nor a preventive treatment (at 2.5 months, before decline) could rescue the cognitive decline in this high expression tau line (Figure 5, Figure 6 and Figure 7). It appears that the very high Tau^P301L^ aggregation propensity in this line overwhelms the protective actions of the drug. Different behavioral paradigms were used to check cognitive performance in rTg4510 mice after rolofylline treatment, but no reversal effects were observed in any of the tests, except a slight improvement in the 12 h nesting test in the therapeutic treatment. In fact, when rTg4510 mice are treated with rolofylline at later stages (5 months old), they display hyperactivity and reduced nesting behavior. At 5 months of age, rTg4510 mice present extensive neuronal death (Appendix A) which could induce an altered distribution or activity of A1 receptors. Since blockade of A1 receptors has an antidepressant effect on neuronal activity, changes in the expression of A1R during the course of the pathology could explain the increase in locomotor activity observed in rolofylline-treated rTg4510 mice in the open field test.

A similar lack of improvement also holds for tau pathology markers such as PHF1, MC1, or Gallyas silver staining, all of which correlate with the failure of reversal of cognitive performance in this mouse line (Figure 6). Notably, the presence of tau tangles in the frontal cortex and olfactory bulb of rTg4510 mice, judged using PET imaging with the tracer [^18^F]PI-2620 and confirmed by immunohistochemistry and biochemistry, is much higher than in mice accumulating Tau^ΔK^ or TauRD^ΔK^ (Appendix A). Indeed, the olfactory dysfunction is among the earliest features of AD and FTD [51,52], and the accumulation of aggregated tau protein in this brain structure correlates with the progression and the severity of the disease in other brain regions [53], underscoring the potential utility of olfactory tissue in the early diagnosis of AD and FTD. Moreover, a significantly higher level of inflammation was observed in several brain regions of rTg4510 mice by PET imaging using the tracer [^18^F]DPA-714, compared to the TauRD^ΔK^ mice that showed neuroinflammation at later time points [26] (Appendix A). Although preventive rolofylline treatment was started as early as 2.5 months of age in the rTg4510 mice, this line presents impairment in the MWM as early as 1.3 months of age [8], indicating that the initial trigger for cognitive decline occurs at a very early age, so it becomes irreversible despite rolofylline treatment. Nevertheless, we cannot rule out that an earlier intervention with rolofylline could be also beneficial in this transgenic mouse line. Tau^ΔK^ and TauRD^ΔK^ lines showed a milder and more progressive tau pathology when compared to the rTg4510 mouse line. Tau^ΔK^ and TauRD^ΔK^ lines have a low rate of neuronal death which occurs only at late time points when the synaptopathology and cognitive decline are present, and they have a normal life span. In contrast, rTg4510 mice present cognitive deficits, loss of body weight, and tau pathology very early. Therefore, it appears that the beneficial effect of rolofylline seen in the Tau^ΔK^ and TauRD^ΔK^ lines can be overwhelmed when the expression of mutant tau becomes excessive, as in the rTg4510 line.

On the other hand, several inconsistencies regarding metabolic activities (as determined by [^18^F]FDG PET) have been reported in the different AD transgenic mouse lines, depending on the type of pathology, age, and treatment of the animals [18]. This might also affect different effects of rolofylline in the rTg4510 mice. However, the main source of unexpected observations on the rTg4510 mouse line appears to be its several genetic changes in addition to that of tau. This has been discovered only recently [35], much later than the early publications on the mouse line (starting from [9]. Over the years, authors have noted changes in pathological markers or behavior that could be explained in terms of the high level of mutant tau, whereas others could not, notably the aspect of hyperactivity and high level of inflammation (e.g., [8,54]). It is therefore not surprising that a drug like rolofylline, affecting neuronal metabolism, yields atypical results not seen in other tau mouse models.

### 3.3. Synaptic Effects vs. Rolofylline Treatment

With regard to synaptic effects, expression of pro-aggregant Tau^ΔK^ and TauRD^ΔK^ caused a profound synaptic decay and loss of dendritic spines [11,29]. Synaptic degeneration precedes neuronal loss and best correlates with tau pathology and cognitive performance in dementing illnesses [55,56]. Several AD mouse models present early synaptic failure [57,58,59] and reduction or regression in dendritic spines [60,61,62]. Treatment with rolofylline restores the dendritic spine level in organotypic hippocampal slices from Tau^ΔK^ mice and normalizes basal synaptic transmission [22]. Indeed, rolofylline treatment increases the expression of the synaptic protein PSD-95 in the hippocampus of Tau^ΔK^ and TauRD^ΔK^ together with a recovery of the number of spines, as analyzed by Golgi staining in the Tau^ΔK^ mice. Changes in the levels of PSD-95 are likely to induce alterations in synaptic signaling and plasticity, which in turn leads to learning impairments [63]. Therefore, amelioration of PSD-95 protein deficits could be related, in part, to the rescue of memory deficits in rolofylline-treated Tau^ΔK^ and TauRD^ΔK^.

Refining treatment paradigms with respect to treatment initiation, concentration, and duration together with early detection of the pathology are key steps in the reversal or delay of disease progression. Therefore, selection of the right time point and window of intervention is crucial for each treatment to reduce tau pathology. A clear example is the methylene blue (MB) treatment, a drug that inhibits tau aggregation. MB therapeutic treatment (after the onset of cognitive impairment) failed to avert or recover learning and memory deficits in Tau^ΔK^ and TauRD^ΔK^. Although A1R inhibition was not statistically significant in cognitively relevant brain areas (e.g., hippocampus and frontal cortex) in our PET study, rolofylline-induced remodeling of functional networks may lead to higher neuronal activity in those brain regions as well. This hypothesis will be addressed in further studies. In contrast, preventive MB application starting before onset of functional deficits preserved cognition in Tau^ΔK^. Moreover, preventive MB application starting at a young age showed a greater impact on cognitive performance, reflecting the importance of the treatment design in order to assess the drug efficacy [64].

The adenosine A_1_ receptor is a G_i_/G_0_-protein-coupled receptor that inhibits adenylate cyclase and reduces cAMP levels [20]. Rolofylline treatment could produce an increase in the levels of cAMP, inducing the activation of protein kinase A [65], which in turn promotes the activation of the ubiquitin-proteasome system [66,67] that plays a central role in the clearance of unfolded proteins [68]. Although these pathways need to be further investigated, they could play a role in the reduction of aberrant phosphorylated and misfolded tau observed in the Tau^ΔK^ mice. Consistent with this, several studies revealed that drugs promoting increased cAMP levels have a beneficial effect in different neurodegenerative diseases, e.g., AD, Huntington’s and Parkinson’s disease which are associated with tauopathy [69,70,71]. This makes drugs targeting adenosine A_1_ receptors promising for the treatment/prevention of neurodegeneration, independently of drugs targeting other adenosine receptors such as A2A receptors (also alleviating tau pathology [72]).

## 4. Materials and Methods

### 4.1. Animals and Rolofylline Treatment

All animal experiments were carried out in accordance with the guidelines of the German Welfare Act and approved by the local authorities (Landesamt für Natur, Umwelt und Verbraucherschutz Nordrhein-Westfalen, Recklinghausen, Germany). Animals were housed in groups of 2–5 animals under standard conditions (23 °C, 40–50% humidity, ad libitum access to food and water) with a 12 h light/dark cycle (with light on from 6 a.m. to 6 p.m.).

The transgenic mouse models included the following: (a) rTg(tauP301L)4510 [8,9]; (b) transgenic mice expressing full-length pro-aggregant human tau (2N4R Tau-ΔK280, termed Tau^ΔK^) [38]; or (c) its repeat domain (TauRD-ΔK280, TauRD^ΔK^) [26]. Briefly, rTg4510 animals were produced by crossing the activator mouse line CamKII-tTA with the responder tetO.MAPT*P301L mouse line. Mice having both CaMKII-tTA and tau transgene expressed human mutant P301L tau (0N4R). Transgenic mice expressing pro-aggregant human full-length tau (2N4R, Tau441, with deletion mutant ΔK280, 441 − 1  =  440 amino acids, here termed Tau^ΔK^) or pro-aggregant tau repeat domain (construct K18, 4R, residues 244–372, termed TauRD^ΔK^) were generated as described [26,38]. Briefly, responder mice carrying either the tau-transgene together with a luciferase reporter were crossbred with the CaMKIIα-tTA transactivator mice [73] to obtain double-transgenic mice with constitutive expression of luciferase and pro-aggregant Tau^ΔK^ or TauRD^ΔK^. All bigenic offspring were heterozygous and had an identical C57BL/6 genetic background. Non-transgenic littermates were used as controls. The transgene expression of bigenic mice started roughly around birth (0mo) concomitant with the onset of CaMKIIα activity. The expression was measured in vivo using bioluminescence imaging of luciferase activity [11].

For oral application of rolofylline, mice were treated with custom-made food pellets (Ssniff Spezialdiäten GmbH, Soest, Germany) containing 1.5 mg adenosine A_1_ receptor antagonist rolofylline (KW-3902) per kg. The dosage of rolofylline was 0.225 mg·kg^−1^·d^−1^. The drug dose was calculated considering an average body weight of 30 g per mouse and a daily food intake of 4.5 g per mouse and day [74]. The mouse dose of 0.225 mg·kg^−1^·d^−1^ relates to a phase III study published by [43] (NCT00354458) that corresponds to a daily dose of 0.5 mg·kg^−1^·d^−1^ in humans and to the fact that species-dependent differences suggest a ~28-fold stronger binding affinity of rolofylline to the rodent A1 receptor in comparison to the human A1 receptor (KI_human_ = 5 nM, [75]; KI_rodent_ = 0.19 nM, [76]. Nine to sixteen control littermates and transgenic mice were treated with rolofylline at different ages for 2 weeks, and then behavior analysis was started. Animals were continuously treated until the end of the behavioral study. Dosage was equal for all groups.

### 4.2. Histology and Golgi Staining

Immunohistochemistry was performed on 5 μm paraffin sections as described [11,29]. The following antibodies were used for light microscopy: MC1 (conformational epitope, aa 5–15  +  312–322, 1:10, gift of Dr. P. Davies, Albert Einstein College, NY, USA); phospho-tau antibodies 12E8 (pSer262/pSer356, 1:2000, Elan, Los Angeles, CA, USA); AT8 (pSer202/pThr205, 1:500, Thermo Scientific, Waltham, MA, USA); Alz-50 and PHF-1 (pS396/pS404) (1:50, were a kind gift from Dr. P. Davies); NeuN (1:1000, Millipore, Burlington, MA, USA); Iba1 (1:1000, Wako, Neuss, Germany); and GFAP (1:2000, Sigma, St. Louis, MO, USA). Secondary antibodies as well as the avidin-biotinylated peroxidase complex were provided by the Vectastain Universal Elite ABC kit (Vector Laboratories, Newark, CA, USA), and DAB (Dako, Carpinteria, CA, USA) was used to visualize the antibody labeling. Gallyas silver impregnation was performed as described [11,29].

For the Golgi-Cox impregnation of neurons [77], the FD rapid GolgiStain TM kit (FD NeuroTechnologies, Columbia, MD, USA) was used according to the manufacturer’s protocol. A total of 120 μm floating sections of transgenic and WT mice at ~17.5 months of age were Golgi-impregnated, and hippocampal pyramidal CA1-neurons were used for the quantification of dendritic spines as described [62]. For each mouse (*n*  =  3–4 per group), three different brain slices were taken, and spines of ~10 neurons per slice were counted. For each neuron, 1–2 secondary dendrites of 20–30 μm lengths were quantified using ImageJ software (NIH). Stainings were imaged using a LSM510 Meta confocal microscope (Zeiss, Oberkochen, Germany).

Histology and golgi staining results are summarized in Table 1.

### 4.3. Biochemical Analysis of Brain Tissue

Sarkosyl-extraction, total protein preparation, and western blots were performed as described previously [11,29]. Depending on the first antibody, 2–20 μg of total protein or 3 μL of sarcosyl extraction lysates from brain tissue were loaded for the detection with pan-tau antibody K9JA (1:20000, Dako A-0024), phospho-tau antibodies 12E8 (pSer262/pSer356, 1:500, Elan), and PSD-95 (1:1000, ThermoFischer). Blots were normalized through the concentration of β-actin (1:20,000, Sigma) or β-III-tubulin (1:5000, RD Systems, Minneapolis, MN, USA), visualized with ECL Plus detection system (GE Healthcare, Chicago, IL, USA) and quantified using a computer-assisted densitometer (Gel-Pro Analyzer, version 4; Media Cybernetics, Bethesda, MD, USA).

Biochemical results are summarized in Table 1.

### 4.4. Behavioral Tests

Nesting test. Mice were single-housed with one nestlet per cage. The nestlet was removed after 2, 6 and 12 h, weighted, and the percentage of nesting was assessed at each time point (% nesting = 100 − ((final weight)/(initial weight)) × 100). Nest-building scores were defined as previously described [30].

Burrowing test. The burrowing test was performed as described elsewhere [30]. The final food pellets’ weight burrowed overnight was measured the next morning and the percentage of burrowing determined (% burrowing = 100 − ((final weight)/(initial weight)) × 100).

Fear conditioning. Mice were tested in the fear conditioning apparatus as described elsewhere [46] Fear conditioning was conducted in transparent plastic boxes (21.5 × 20 × 25 cm) with stainless-steel grid floors connected to an aversive stimulator (Med Associates). Briefly, during the training, mice were introduced into the chamber and allowed to explore it for 2 min. A tone (sound cue) was then presented at a level of 80 dB for 30 s. A mild foot shock (0.5 mA) was administered during the last 2 s of the tone presentation and co-terminated with the tone. After the shock presentation, mice were kept for 30 s in the chamber to allow memory consolidation. Contextual and cued fear conditionings were tested 24 h after the training. For contextual conditioning test, mice were placed into the chamber, and the percentage of freezing per minute was analyzed during 4 min. After 4 min, the cue conditioning test was performed by presenting the tone for 30 s. Freezing in the presence and after the tone was analyzed over 2 min and compared to analyze cue-induced memory. Results were expressed as the mean percentage of freezing in the presence and absence of tone.

Morris water maze (MWM). Spatial memory abilities were examined in the standard hidden-platform acquisition and retention version of the MWM [47].

A 2-day pre-training protocol was conducted to familiarize the mice with swimming and climbing to a hidden platform (22 °C water temperature, 4 trials/day, maximum trial duration 60 s, 20 min inter-trial interval). The pre-training platform (10 cm diameter) was placed 1 cm below the water’s surface.

In the learning and probe trial phase, a 150 cm circular pool was filled with water opacified with nontoxic white paint (Biofa Primasol 3011, Müllheim, Germany) and kept at 22 °C, as previously described [78]. In the middle of the target quadrant, a 15 cm round platform was hidden (1 cm beneath the water surface). The MWM room was equipped with visual cues to facilitate orientation. The pool was divided into four quadrants: target (T), right adjacent (R), opposite (O), and left adjacent (L). Each mouse performed four swimming trials per day (maximum duration 60 s, 15–20 min inter-trial interval) for five consecutive days. Mice started the test from six symmetrical positions in a pseudo-randomized order across trials. When mice failed to find the submerged platform within 60 s, they were guided to the platform, remaining there for 10 s before returning to their cage. The escape latency was determined. On acquisition days 3, 4, 5, and 3 days after the end of the acquisition phase (day 8), a probe trial was conducted by removing the platform and recording the search pattern of the mice for 60 s. The following three to four learning trials were carried out with the platform placed in the initial position on the target quadrant to avoid extinction. During the acquisition and probe trials, the EthoVision XT version 8.5 video tracking system was used (Noldus Information Technology, Wageningen, The Netherlands).

Open field and novel object recognition test (NORT). Mice were tested in a square open field (45 cm long) (Panlab, Barcelona, Spain) located in a room with dim lighting. NORT was performed as previously described [79]. Briefly, mice were habituated to the open field in the absence of the objects for 10 min/d over 2 d. During the training period, mice were placed in the open field with two identical objects for 10 min. The retention test was performed 24 h post-training (long-term memory) by placing the mice back to the open field for a 5-min session and by randomly exchanging the familiar object for a novel one. Data were recorded and analyzed using EthoVision XT video tracking system version 8.5 (Noldus Information Technology).

Y-maze. The Y-maze task was used to analyze hippocampus-dependent memory in rolofylline-treated mice. The dimensions of the used Y-maze were 30 × 6 × 15 cm (length by width by height) (Panlab). In the training session, one arm was closed (novel arm), and mice were placed in the stem arm of the Y (home arm) and allowed to explore this arm and the other available arm (familiar arm) for 10 min. The mice were placed back in their home cage after exploration. To assess long-term memory, 4 h post-exploration, the closed arm was opened, and mice were placed in the stem arm of the Y-maze and allowed to freely explore all three arms for 5 min. Arm preference was determined by calculating the following: time spent in each individual arm*100/total time spent in both arms (both familiar and novel arm).

Behavioral experiments are summarized in Table 1.

### 4.5. Positron Emission Tomography (PET)

In order to ascertain that systemically administered rolofylline reached the brain and engaged with the drug target, PET measurements were performed with six control mice at the age of seven months. Each animal received one measurement with 10 MBq [^18^F]CPFPX alone (a selective A1 receptor ligand), and one measurement with 10 MBq [^18^F]CPFPX + 10 mg/kg rolofylline, on two separate days. Animals were anesthetized (initial dosage: 5% isoflurane in O_2_/air 3:7, then reduction to 2%), and a catheter for tracer injection was inserted into the lateral tail vein. Mice were placed on an animal holder (medres^®^ GmbH, Cologne, Germany) and fixed with a tooth bar in a respiratory mask. Body temperature was maintained at 37 °C using a feedback-controlled warming system (medres^®^). Eyes were protected from drying with Bepanthen eye and nose ointment (Bayer). A PET scan in list mode was conducted using a Focus 220 micro PET scanner (CTI-Siemens, Erlangen, Germany) with a resolution at the center of field of view of 1.4 mm. Data acquisition started with intravenous tracer injection (volume: 125 µL) and lasted for 30 min. This was followed by a 10 min transmission scan using a ^57^Co point source for attenuation correction. After the scan was finished, the catheter was removed, and the mouse was returned to its home cage. To visualize tau accumulation and inflammation, the tracers [^18^F]PI-2620 and [^18^F]DPA-714 were administered to rTg4510 mice (*n* = 8 rTg4510 and 8 non-transgenic littermates) and TauRD^ΔK^ mice (*n* = 9 TauRD^ΔK^ and 7 non-transgenic littermates) as described above with scan durations of 1 h ([^18^F]PI-2620) and 30 min ([^18^F]DPA-714).

After full 3D rebinning, summed images (0–30 min for [^18^F]CPFPX and [^18^F]DPA-714, 30–60 min for [^18^F]PI-2620) were reconstructed using the iterative OSEM3D/MAP procedure resulting in voxel sizes of 0.47 × 0.47 × 0.80 mm. For all further processing of the images including statistics, the software VINCI 4.72 for MacOS X (Max Planck Institute for Metabolism Research, Cologne, Germany) was used. Images were first co-registered and then intensity-normalized to background. For this, a spherical volume of interest (VOI) with a volume of 4 mm^3^ was placed inside the neck musculature immediately caudal to the cerebellum ([^18^F]CPFPX) or into the midbrain ([^18^F]PI-2620 and [^18^F]DPA-714). Each image was divided by the mean value of the background VOI, resulting in the “background standardised uptake value ratio” (SUVR_bg_).

The [^18^F]CPFPX images with and without rolofylline were compared voxel-wise using the paired *t*-test. This was followed by a threshold-free cluster enhancement (TFCE) procedure with subsequent permutation testing [80], resulting in a statistical map corrected for multiple testing, thresholded at *p* < 0.05. Because TFCE values are used for final thresholding only, the color bars of TFCE maps were labeled with the original t-values, marked t_TFCE_. In addition, VOIs were drawn for different brain areas and VOI mean values were extracted. They were used for paired *t*-tests in Graphpad Prism 6.0 for MacOS X. Significance level was always *p* < 0.05.

**Table 1 ijms-24-09260-t001:** Summary of all behavioral and biochemical/histological results for all mouse lines. All the changes are present compared to LCtrl, and the number of animals per experiment is mentioned. Some animals were excluded in some experiments due to problems in performing. ↑ and ↓, increased or decreased compared to LCtrl; ++, higher expression compared to LCtrl; +, slightly higher expression compared to LCtrls; =, not statistically different compared to LCtrl; N/A, not applicable (experiment not performed in this mouse line); NORT, novel object recognition test; FC, fear conditioning; MWM, Morris water maze; *, experiment performed on [22].

										Therapeutic Treatment				Preventive Treatment			
		LCtrl	LCtrl Rol	Tau^∆K^	Tau^∆K^ Rol	LCtrl	LCtrl Rol	TauRD^∆K^	TauRD^∆K^ Rol	LCtrl	LCtrl Rol	rTg4510	rTg4510 Rol	LCtrl	LCtrl Rol	rTg4510	rTg4510 Rol
**Behavior experiments**	**Y-maze**	-	-	↓*	↑*	N/A	N/A	N/A	N/A	(n = 14)	(n = 14)	↓ (n = 9)	↓ (n = 9)	N/A	N/A	N/A	N/A
	**NORT**	-	-	↓*	↑*	N/A	N/A	N/A	N/A	N/A	N/A	N/A	N/A	N/A	N/A	N/A	N/A
	**FC**	-	-	↓*	↑*	(n = 9)	(n = 11)	↓ (n = 10)	↑ (n = 12)	N/A	N/A	N/A	N/A	N/A	N/A	N/A	N/A
	**Body weight**	**(n = 7)**	**(n = 9)**	´=(n = 9)	´=(n = 8)	(n = 9)	(n = 11)	↓ (n = 10)	↑ (n = 13)	(n = 12)	(n = 14)	↓ (n = 10)	↓ (n = 10)	N/A	N/A	N/A	N/A
	**Burrowing test**	N/A	N/A	N/A	N/A	(n = 9)	(n = 10)	↓ (n = 10)	↑ (n = 13)	(n = 14)	(n = 14)	↓ (n = 9)	↓ (n = 10)	N/A	N/A	N/A	N/A
	**Learning MWM**	N/A	N/A	N/A	N/A	(n = 9)	(n = 11)	↓ (n = 10)	↑ (n = 13)	N/A	N/A	N/A	N/A	(n = 16)	(n = 9)	↓ (n = 13)	↓ (n = 13)
	**LTPT MWM**	N/A	N/A	N/A	N/A	N/A	N/A	N/A	N/A	N/A	N/A	N/A	N/A	(n = 16)	(n = 9)	´=(n = 13)	´=(n = 13)
	**Nesting test**	N/A	N/A	N/A	N/A	N/A	N/A	N/A	N/A	(n = 14)	(n = 14)	↓ (n = 9)	↓↓ (n = 9)	(n = 16)	(n = 9)	↓ (n = 13)	↓ (n = 13) (12 h not different than LCtrl)
	**OF**	N/A	N/A	N/A	N/A	(n = 9)	(n = 11)	´=(n = 10)	´=(n = 13)	(n = 14)	(n = 14)	´=(n = 9)	↑ (n = 9) (hyperactivity)	(n = 16)	(n = 9)	´=(n = 13)	´=(n = 13)
	**OF time in center**	N/A	N/A	N/A	N/A	N/A	N/A	N/A	N/A	N/A	N/A	N/A	N/A	(n = 16)	(n = 9)	´=(n = 13)	↓ (n = 13) (anxiety)
**Taupathology**	**Gallyas silver staining**	(n =4)	(n = 4)	++ (n = 4)	+ (n = 4)	(n = 4)	(n = 4)	++ (n = 4)	+ (n = 4)	(n = 4)	(n = 4)	++ (n = 4)	++ (n = 4)	N/A	N/A	N/A	N/A
	**pTau12E8**	N/A	N/A	↑ (n = 6)	↓ (n = 8)	(n = 4)	(n = 4)	++ (n = 4)	+ (n = 4)	N/A	N/A	N/A	N/A	N/A	N/A	N/A	N/A
	**Alz50**	(n = 4)	(n = 4)	↑ (n = 4)	↓ (n = 4)	N/A	N/A	N/A	N/A	N/A	N/A	N/A	N/A	N/A	N/A	N/A	N/A
	**PHF1**	N/A	N/A	N/A	N/A	N/A	N/A	N/A	N/A	(n = 4)	(n = 4)	++ (n = 4)	++ (n = 4)	N/A	N/A	N/A	N/A
	**MC1**	N/A	N/A	N/A	N/A	N/A	N/A	N/A	N/A	(n = 4)	(n = 4)	++ (n = 4)	++ (n = 4)	N/A	N/A	N/A	N/A
**Synaptic markers**	**PSD-95**	**(n = 6)**	**(n = 6)**	↓ (n = 6)	↑ (n = 8)	N/A	N/A	N/A	N/A	N/A	N/A	N/A	N/A	N/A	N/A	N/A	N/A
	**Spines**	**(n = 2)**	**(n = 4)**	↓ (n = 3)	↑ (n = 3)	N/A	N/A	N/A	N/A	N/A	N/A	N/A	N/A	N/A	N/A	N/A	N/A
	**Brain weight/ HIP Volume**	**(n = 5)**	**(n = 5)**	´=(n = 6)	´=(n = 6)	(n = 9)	(n = 10)	↓ (n = 10)	↓ (n = 12)	(n = 13)	(n = 13)	↓ (n = 8)	↓ (n = 9)	(n = 12)	(n = 9)	↓ (n = 7)	↓ (n = 10)

## 5. Conclusions

There are currently no drugs to reverse AD in humans, and only a few that slow down disease progression with limited efficacy. By contrast, on the level of experimental mouse models of AD-like tauopathy, it is possible not only to slow down but also to halt or reverse symptoms such as cognitive decline. Examples are the switch-off of mutant tau expression (in case of regulatable gene expression [9,38], tau aggregation inhibitors (e.g., methylene blue or derivatives that halt progression [5,64]), or compounds that affect tau only indirectly (e.g., adenosine receptors [22,72]). In the latter cases, the signaling pathways affecting tau are not well understood; however, known tau-interactors point to presynaptic or mitochondrial components [48,56]. As shown here for the case of A1 antagonists, the curative potential holds for different proaggregant mutations of tau and for different variants of tau, as long as the level of tau remains below a critical limit.

## Figures and Tables

**Figure 1 ijms-24-09260-f001:**
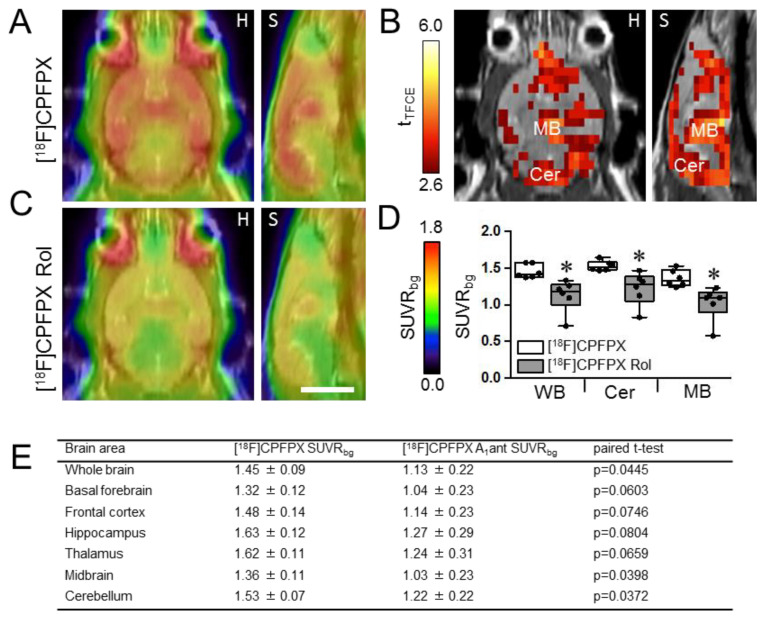
PET imaging: Blocking of [^18^F]CPFPX uptake with rolofylline in brains of WT mice. Animals were injected intravenously with 10 mg/kg rolofylline together with 10 MBq [^18^F]CPFPX, a tracer that binds specifically to adenosine A1 receptors. (**A**,**C**) Average images of [^18^F]CPFPX SUVR_bg_ in horizontal (H) and sagittal (S) view, projected onto an MRI template in the absence of rolofylline (**A**) or injected with [^18^F]CPFPX + 10 mg/kg rolofylline (**C**). (**B**) Significant differences on the voxel level between the two measurements, analyzed using paired *t*-test corrected for multiple testing (*p* < 0.05). (**D**) Box plots of VOI values. * *p* < 0.05 when comparing to unblocked [^18^F]CPFPX uptake. (**E**) Analysis of the [^18^F]CPFPX SUVR_bg_ in different brain areas. Results are expressed as mean ± SD (*n* = 6/group). Abbreviations: Cer: cerebellum; MB: midbrain; WB: whole brain. Scale bar: 5 mm.

**Figure 2 ijms-24-09260-f002:**
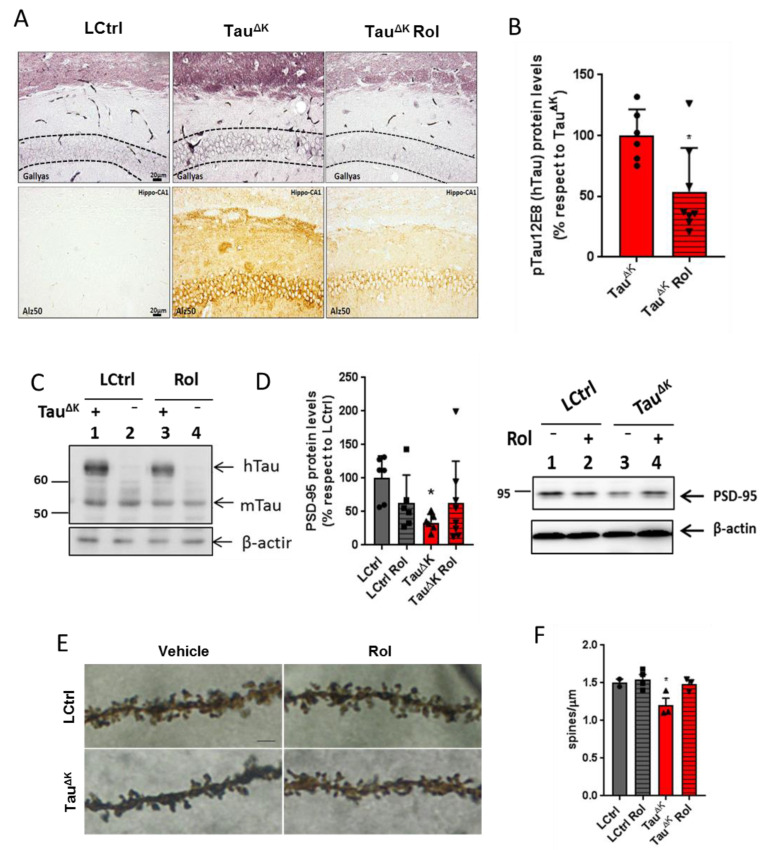
Rolofylline decreased hippocampal tau pathology in Tau^ΔK^ transgenic mice. Treatment with rolofylline was started at ~15 months, ~3 months after onset of cognitive decline. At age ~17.5 months (after 2.5 months of treatment), brain tissue was obtained for western blotting and histological analysis. For all figures, data are expressed as a mean ± SD. (**A**) Gallyas silver staining of CA1 hippocampal region (upper panels) and Alz50 antibody staining (lower panels). Note the reduction in both pathological markers after rolofylline treatment. Dashed lines mark the region of interest in the Gallyas staining corresponding to the CA1 cellular layer. Scale bar: 20 µm. (**B**) The protein levels of phosphorylated human tau (antibody 12E8; pSer262/pSer356) in the hippocampus of rolofylline-treated Tau^ΔK^ mice were significantly lower compared to the untreated group (*n* = 6–8 per group; Student’s *t*-test; *p* = 0.0204). * *p* < 0.05 when comparing to Tau^ΔK^ mice. (**C**) Example of western blotting with identified bands of human tau (hTau), murine tau (mTau), and loading control β-actin. (**D**) Protein levels of PSD-95 in the hippocampus of untreated Tau^ΔK^ mice were significantly lower when compared to LCtrl (one-way ANOVA, F(3, 21) = 3.23, *p* = 0.0429; Dunnett’s post-hoc *p* = 0.0202; *n* = 6–7/group). * *p* < 0.05 when comparing to LCtrl mice. (**E**) High magnification of CA1 apical dendritic spines from TauRD^ΔK^ and LCtrl mice in the presence or absence of rolofylline treatment. Scale bar: 5 µm. (**F**) Quantification of (**E**). Rolofylline treatment recovers the number of spines in transgenic Tau^ΔK^ animals. There was a significant difference between LCtrl and Tau^ΔK^ mice (*n* = 2–4 per group; F(3, 8) = 5.09, *p* = 0.0293; Tukey’s post-hoc *p* = 0.0254), but not between LCtrl and Tau^ΔK^ Rol mice (*p* = 0.9991). The comparison Tau^ΔK^ versus Tau^ΔK^ Rol revealed a statistic tendency (*p* = 0.0843). * *p* < 0.05 when comparing to LCtrl mice.

**Figure 3 ijms-24-09260-f003:**
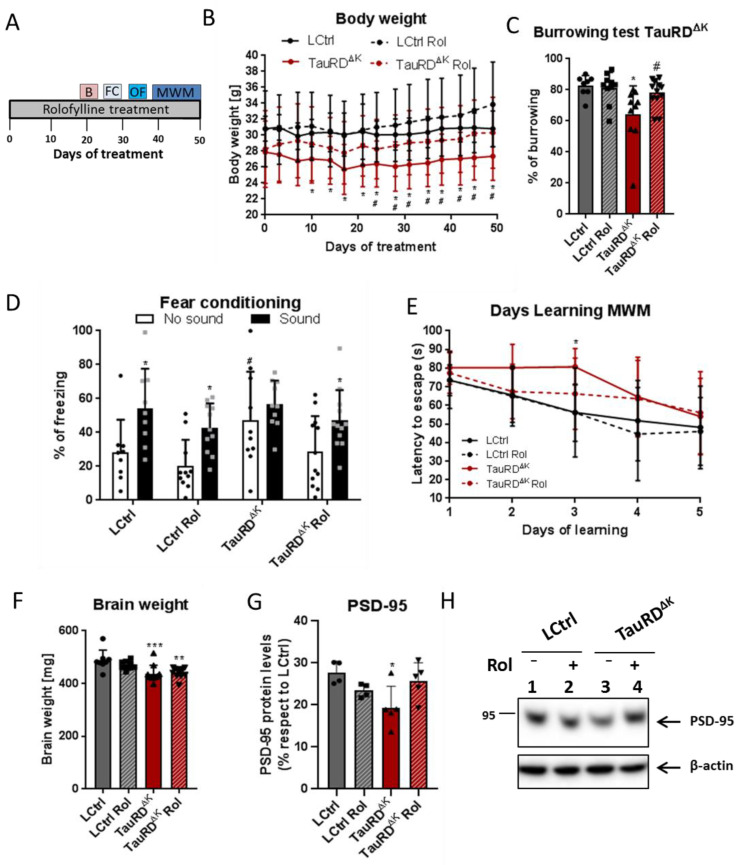
Rolofylline decreased cognitive deficits in TauRD^ΔK^ mice. Fifteen-month-old animals expressing TauRD^ΔK^ and their littermate controls were treated with rolofylline for 2.5 months, ~6 months after onset of cognitive decline. Data are expressed as a percentage mean ± SD. (**A**) Timeline of the behavior paradigms used during the treatment. (**B**) Rolofylline treatment prevented a decrease in body weight observed in animals expressing mutant TauRD^ΔK^. Two-way ANOVA (*n* = 9–13 per group) revealed significant main effects of the factors of “time” (F(14, 546) = 11.79, *p* < 0.0001) and “treatment” (F(3, 39) = 3.27, *p* = 0.0314), as well as a significant factor interaction (F(42, 546) = 2.34, *p* < 0.0001). The latter indicates that body weight developed differently within groups. The body weight of LCtrl mice remained constant throughout the observation period. In contrast, the body weight of TauRD^ΔK^ mice was significantly different from LCtrl from day 10 onwards (Tukey’s post-hoc; * *p* < 0.05) and from rolofylline-treated LCtrls from day 24 onwards (Tukey’s post-hoc; # *p* < 0.05). Rolofylline-treated TauRD^ΔK^ Rol mice showed no statistical difference compared to control animals at any time point analyzed. (**C**) After 20 days of rolofylline treatment, TauRD^ΔK^ mice showed improved burrowing behavior (F(3, 36) = 4.87, *p* = 0.0061; *n* = 8–12 per group). There was a significant difference between LCtrl and TauRD^ΔK^ (Tukey’s post-hoc *p* = 0.0125; * *p* < 0.05 when compared to LCtrl) and between TauRD^ΔK^ and TauRD^ΔK^ Rol (*p* = 0.0471; # *p* < 0.05 when compared to TauRD^ΔK^). LCtrl and TauRD^ΔK^ Rol were not significantly different with respect to burrowing (*p* = 0.8368). (**D**) Rolofylline treatment also improved performance in the fear conditioning test. There was a significant difference between contextual (no sound) and cue-induced (sound) freezing for LCtrl and LCtrl Rol (F(1, 76) = 19.62, *p* < 0.0001, Sidak’s post-hoc *p* < 0.05, black *), but not for TauRD^ΔK^ mice (*p* = 0.7525). Rolofylline-treated TauRD^ΔK^ Rol mice showed a statistic tendency (*p* = 0.0905, grey *). TauRD^ΔK^ mice showed more contextual freezing (no sound) than the other groups, which was significant compared to the LCtrl Rol group (F(3, 76) = 3.99, *p* = 0.0108, Sidak’s post-hoc *p* = 0.0138; #). (**E**) Rolofylline treatment improved learning speed of TauRD^ΔK^ mice in the Morris water maze test (Two-way ANOVA F(3, 39) = 3.56, *p* = 0.0227 for factor “treatment”, *n* = 9–13 per group). TauRD^ΔK^ mice needed more time to decrease the latency to escape, with a significant difference between LCtrl and TauRD^ΔK^ on learning day 3 (Dunnett’s post-hoc *p* = 0.0084, *). In contrast, there was no significant difference between LCtrl and TauRD^ΔK^ Rol (*p* > 0.28 on all days tested). (**F**) The loss in brain weight in TauRD^ΔK^ animals was not reversed by rolofylline treatment (F(3, 36) = 7.38, *p* = 0.0006, *n* = 8–12 per group). Both TauRD^ΔK^ and TauRD^ΔK^ Rol mice had significantly lighter brains compared to LCtrl (Dunnett’s post-hoc *p* = 0.0006 *** and 0.0022 **, respectively). (**G**) Protein levels of PSD-95 in the hippocampus of untreated TauRD^ΔK^ mice were significantly lower when compared to LCtrl (one-way ANOVA, F(3, 14) = 4, *p* = 0.0285; Tukey’s post-hoc *p* = 0.0263, *; *n* = 4–5/group); meanwhile, no differences were observed between LCtrl and rolofylline-treated TauRD^ΔK^ (*p* = 0.8637). There was no difference between transgenic animals, although PSD-95 had a clear trend to increase after rolofylline treatment (*p* = 0.0831). (**H**) Representative western blot of PSD-95.

**Figure 4 ijms-24-09260-f004:**
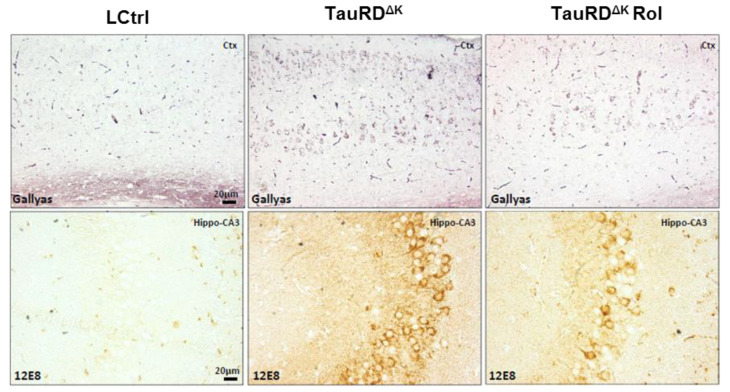
Rolofylline decreases taupathology in TauRD^∆K^ mice. Treatment with rolofylline was started at ~15 months, ~3 months after onset of cognitive decline. At age ~17.5 months (after 2.5 months of treatment), brain tissue was obtained for histological analysis. Gallyas silver staining of cortical region (upper panels) and phosphorylated human tau (antibody 12E8; pSer262/pSer356) antibody staining in the CA3 hippocampal cellular layer (lower panels). Note the reduction in both pathological markers after rolofylline treatment. Scale bar: 20 µm.

**Figure 5 ijms-24-09260-f005:**
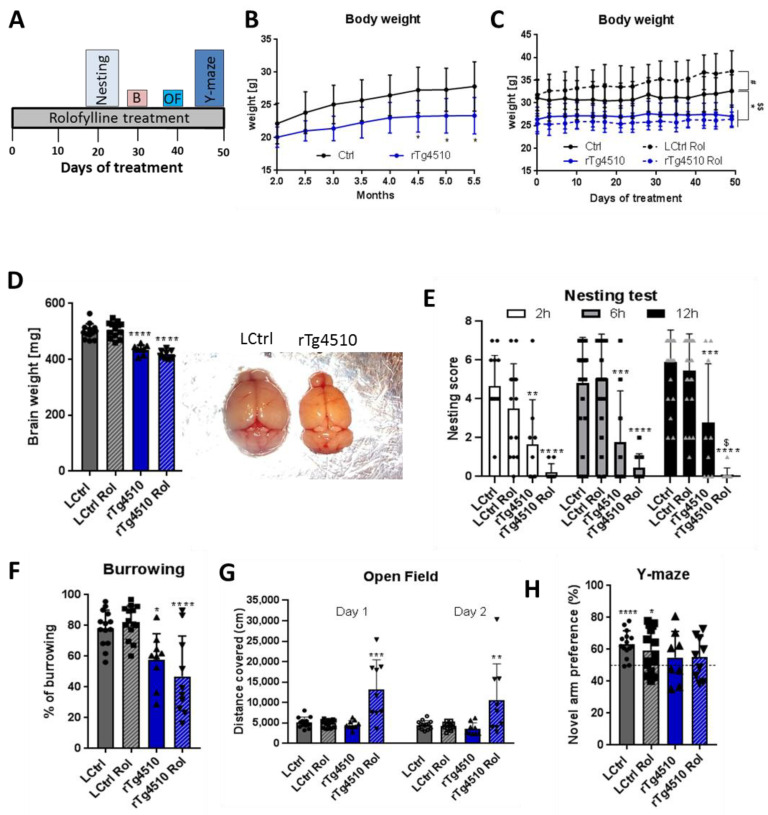
Rolofylline does not improve behavior in rTg4510 mice when administered after cognitive impairment (~5 to 7 months of age): Behavioral assessment. Control and rTg4510 animals were treated with food containing 5 mg/kg of rolofylline during 2 months at ~5 month of age. After 3 weeks of treatment, several behavior assessments were performed. For all experiments, results are expressed as a mean ± SD (*n* = 9–14/group). (**A**) Timeline of the behavior paradigms used during the treatment. (**B**) Body weight was monitored from 2 to 5.5 months of age in rTg4510 mice without rolofylline treatment (blue curve). It was consistently reduced compared to littermate controls (black curve). Two-way ANOVA revealed a significant main effect of the “genotype” factor (F(1, 16) = 11.12; *p* = 0.00042), and Sidak post-hoc comparison showed a significant difference between rTg4510 mice and controls at time points of 4.5–5.5 months of age (*p* < 0.05, *). (**C**) Body weight was monitored every 3–4 days during 50 days of rolofylline treatment (from ~5 to 7 months of age). Rolofylline did not reverse the lower body weight observed in the rTg4510 animals. Two-way ANOVA revealed a significant main effect of the factors of “treatment” (F(3, 432) = 17.83; *p* < 0.0001) and “time” (F(14, 588) = 18.01; *p* < 0.0001) and a significant interaction of the factors of “time” and “treatment” (F(42, 588) = 5.517; *p* < 0.0001), which suggests a different development of body weight between groups. When comparing the mean body weight between groups, Tukey’s post-hoc test showed a significant difference between LCtrl and rTg4510 (*p* < 0.05, *) and rTg4510 Rol throughout (*p* < 0.05, $$), while rTg4510 Rol and rTg4510 were not significantly different (*p* = 0.7898). The body weight of LCtrl Rol was significantly different from LCtrl (*p* < 0.05; #). (**D**) The loss in brain weight in rTg4510 animals was not reversed by rolofylline treatment (F(3, 39) = 38.03, *p* < 0.0001, *n* = 8–13 per group). Both rTg4510 and rTg4510 Rol mice had significantly lighter brains compared to LCtrl (Tukey’s post-hoc *p* < 0.0001). (**E**) Nesting test with rolofylline or without rolofylline treatment for rTg4510 mice and controls, scored after 2, 6, and 12 h after introduction of the nestlet. Transgenic animals showed a worse performance (main effect of “treatment” factor in two-way ANOVA with F(3, 170) = 46.58; *p* < 0.0001; Tukey’s post-hoc *p* < 0.01 at 2 h, *; *p* < 0.001at 6 and 12 h, **; for comparison with LCtrl). Nesting becomes even more impaired by rolofylline treatment, leading to a significant difference between rTg4510 and rTg4510 Rol after 12 h (Tukey’s post-hoc *p* = 0.0284, $). In contrast, treatment has no effect on controls (*p* > 0.44 at all timepoints). (**F**) Burrowing test: % food pellets burrowed overnight. The rTg4510 animals show reduced burrowing behavior, which was not improved by rolofylline treatment (F(3, 43) = 11.84; *p* < 0.0001; Tukey’s post-hoc test showed significant differences between LCtrl and rTg4510 [*p* = 0.0254, *] as well as between LCtrl and rTg4510 Rol [*p* = 0.0002, ***], while there was no significant difference between rTg4510 and rTg4510 Rol [*p* = 0.5054]). (**G**) Open field test: graph showing the covered distance (in cm) in the open field arena during 10 min for 2 consecutive days. Transgenic mice and littermate controls showed similar behavior. Rolofylline treatment had no effect on controls but promoted hyperactive behavior in tau transgenics. Day 1: F(3, 42) = 15.47; *p* < 0.0001; Tukey’s post-hoc *p* < 0.0001 for rTg4510 Rol versus all other groups (***). Day 2: F(3, 42) = 6.26; *p* = 0.0013; Tukey’s post-hoc *p* < 0.01 for rTg4510 Rol versus all other groups (**). (**H**) Y-maze test: The percentage of novel arm exploration time 4 h after the training session of all groups is shown. In the one-sample *t*-test, control and control rolofylline animals had an increased preference for the new arm (*p* < 0.0001 **** and *p* = 0.0192 *, respectively), which was not statistically significant for the transgenic animals (*p* = 4.227 non-treated rTg4510 mice and *p* = 0.2684 for rolofylline-treated rTg4510 mice).

**Figure 6 ijms-24-09260-f006:**
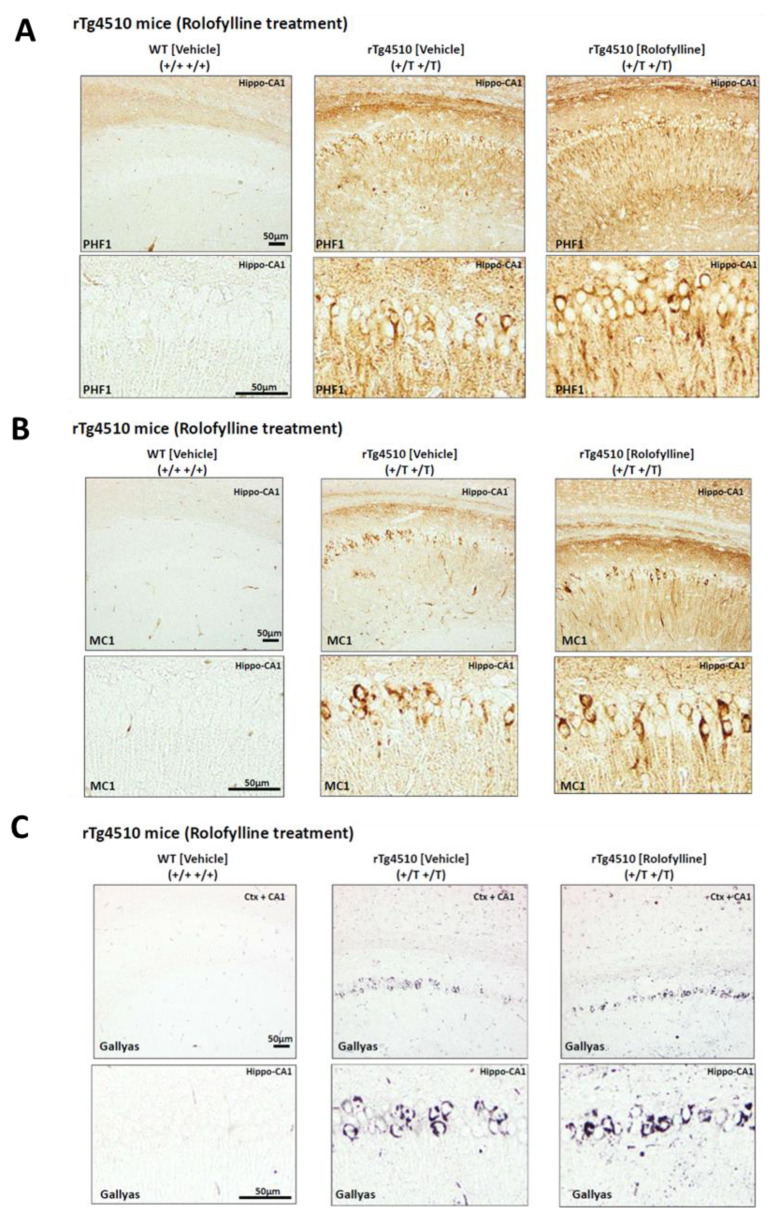
Rolofylline does not pathologically improve phosphorylated and aggregated tau in rTg4510 mice administered from 5 to 7 months of age: Histology. Histological analysis of conformationally changed and phosphorylated tau in the hippocampus (CA1) of rTg4510 mice after ~2 months of rolofylline treatment. In all cases, there was no significant improvement with rolofylline treatment. (**A**) Phosphorylated tau PHF1 antibody (dual phosphorylation epitope Ser396 + Ser404). (**B**) Pathological tau conformation antibody MC1 (epitope 5–15  +  312–322). (**C**) Gallyas silver staining of tau aggregates. Scale bar: 50 µm.

**Figure 7 ijms-24-09260-f007:**
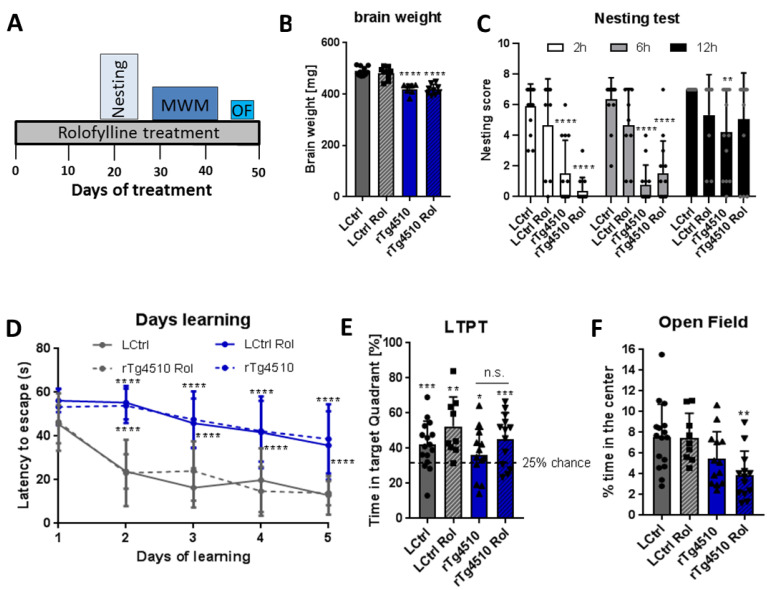
Early (preventive) treatment with rolofylline does not protect against cognitive impairment in rTg4510 mice. Transgenic and control animals were treated with food containing 5 mg/kg of rolofylline at 2.5–3 months of age (at the beginning of the typical onset of tau pathology) for 1.5 months. After 3 weeks of treatment, several behavior assessments were performed. For all experiments, results are shown a mean ± SD (*n* = 7–16/group). (**A**) Timeline of the behavior paradigms used during the treatment. (**B**) Transgenic animals presented reduced brain weight, which was not affected by adenosine A1 antagonist treatment (F(3, 33) = 34.36, *p* < 0.0001, *n* = 7–12 per group). Both rTg4510 and rTg4510 Rol had lighter brains compared to LCtrl (Dunnett’s post-hoc *p* < 0.0001 for both comparisons, ****). (**C**) Nesting test with rolofylline or without rolofylline treatment for rTg4510 mice and controls, scored after 2, 6, and 12 h after introduction of the nestlet. Transgenic animals showed a worse performance at 2 h and 6 h (main effect of the “treatment” factor in two-way ANOVA with F(3, 141) = 44.4; *p* < 0.0001; Tukey’s post-hoc *p* < 0.0001 at 2 h and 6 h, ****; for comparison with LCtrl). At 12 h, non-treated transgenic animals were still performing significantly worse than LCtrl (*p* = 0.0019, **), but rolofylline-treated rTg4510 mice had a clear trend toward improvement (*p* = 0.0569, when compared to non-treated rTg4510 mice). No significant differences were observed in transgenic animals after rolofylline treatment (Tukey’s post-hoc *p* > 0.47 at all points) and treatment has also no effect on controls (*p* > 0.43 at all timepoints). (**D**) The Morris water maze test shows the time (in seconds) mice take to find the hidden platform during the learning phase. Transgenic animals show an increase in latency to reach the platform during the learning trials with no improvement after rolofylline treatment (F(3, 47) = 37.12 for the factor of “treatment”, with Dunnett’s post-hoc *p* < 0.0001 for the comparisons rTg4510 vs. LCtrl and rTg4510 Rol vs. LCtrl at days 2–5, ****). There was no significant difference between rTg4510 and rTg4510 Rol at any time. (**E**) Percentage of time spent in the target quadrant (where the hidden platform was) during the Morris water maze long-term probe trial (LTPT), performed 72 h after the last training day. In the one-sample *t*-test, all groups explored significantly more the platform quadrant (*p* = 0.0001 for LCtrl, ***; *p* = 0.0012 for LCtrl Rol, **; *p* = 0.0138 for rTg4510, *; and *p* = 0.0003 for rTg4510 Rol, ***). When comparing % of exploration time between groups, one-way ANOVA showed no main effect of treatment (F(3, 47) = 2.24, *p* = 0.0964). n.s. = Not Statistically Significant (**F**) Open field test: graph showing the mean percentage of time animals spent in the center of the arena during 10 min. Rolofylline-treated rTg4510 mice presented increased anxiety shown by the reduced time spent in the center of the arena (F(3, 47) = 5.38, *p* = 0.0029, Dunnett’s post-hoc *p* = 0.0023 for the comparison rTg4510 Rol vs. LCtrl, **).

## Data Availability

All relevant data are within the paper and its Appendix A files. Additional datasets during and/or analyzed during the current study are available from the corresponding author on reasonable request.

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
