# Peer review of "Reversal of Tau-Dependent Cognitive Decay by Blocking Adenosine A1 Receptors: Comparison of Transgenic Mouse Models with Different Levels of Tauopathy"

_ijms, 2023, doi:10.3390/ijms24119260_

Round 1

Reviewer 1 Report

The study proposed by Dr. Anglada-Huguet and colleagues addresses the involvement of the adenosine A1 receptor (A1R) in the pathophysiology of tau-associated neurodegeneration and investigates the effects of the A1R antagonist rolophylline in different transgenic models of tauopathy.

Along with previous studies, the Authors have extensively addressed the topic. In essence, the new aspects of the research presented here are: a) the PET results, which show the ability of rolophylline treatment to block the A1R, and b) the lack of the effect of rolophylline on the more aggressive tauopathy mouse model.

Concerning point a), however, the Authors should try to explain the discrepancy between the apparently low (and not statistically relevant) blockade of [18F]CPFPX uptake in brain regions underlying cognition (hippocampus and frontal cortex) and the pro-cognitive effects induced by the treatment. Furthermore, in light of the PET results, which indicate a maximal A1R inhibition of 25% (Fig. 1), the definition of block should be scaled down.

Concerning point b), in addition to the lack of a protective effect of rolophylline in rTg4510 mice, there is clear evidence of a pejorative effect (e.g., Figs. 5E, 5G, 6A, 6B) that should be emphasized and discussed.

Minor points:

- The paragraph on methylene blue in chapter 3 (already present in the discussion) is superfluous.

- Conclusions should be after the Discussion and not after Materials and Methods.

Reviewer 2 Report

Reviewer Comments:

In the manuscript entitled: „ Reversal of tau-dependent cognitive decay by blocking adeno-2 sine A1 receptors: Comparison of transgenic mouse models 3 with different levels of tauopathy”, the authors showed the possibility of novel therapeutic strategy for Alzheimer disease using rolofylline which blocks adenosine A1 receptors in three mouse models expressing different levels of transgene tau, as a whole or truncated molecule, and tau containing a mutation P301L. Although the results for TauP301 mice might be novel, previous findings of this research team already have shown the neuroprotective effect of rolofylline in mouse models manifesting expression of the entire human tau protein molecule or only a truncated form of it in the form of a fragment of repeated microtubule-binding domains. Nevertheless, the work is worth publishing, if only for the new aspects of the analysis.

The manuscript is very carefully prepared from an editorial point of view, nevertheless a few minor errors have crept in, such as in the

Line 764: “images images”

Line 794: “variants variants”

I have no major objections to the reviewed manuscript. The paper is written very well, the description of the Methods is understandable, the Results are clearly presented, and the Discussion is adequate.

A few minor comments the Authors might consider:

(1)   Why is part of the mean data reported with SD and part with SEM?

I always admired researchers for reporting SD.

The European network for the reproducibility of research results has developed a number of guidelines to improve the robustness and repeatability of data. Among other guidance, best practice in statistical analysis was also assessed and it was decided, for example, that SD is much preferable to SEM because it gives a direct measure of variance and is not contaminated by the number of subjects in each group. In addition, it has been decided to move away from bar charts as much as possible because they are not very informative unless the individual data points are also shown as scatter.

Consequently, I am trying to urge the Authors to implement these new guidelines in the current manuscript. I would also appreciate if Authors could convert the bar chart into the bar including individual results.

(2)  Figure 1 shows images from PET in horizontal and sagittal views. Their distinction is obvious to neuroanatomists, but for other recipients of the publication, it would be useful to indicate in the figure which plane of section is which.

(3)  Figure 3 and Figure 5: Why was a different set of behavioral tests used in the tauopathic line presented in Fig.3 and this one presented in Fig.5?

(4)  Figure 5: What are the differences in body weights actually indicative of and does it have anything to do with the treatment of tauopathy?

(5)  Figure 5E: The differences regarding nest building are very intriguing.  hy does Rol harm nest building?

(6)  Figure 7: Similarly, as above, why was a different set of behavioural tests performed in younger and older groups of animals?

(7)  Discussion lines 447 to 450:

The current manuscript does not present data documenting changes in neuronal activity, reduced neuronal ATP levels, decreased mitochondrial density and impaired synaptic function in Tau.K and TauRD.K mice.

(8)  Discussion lines 547 to 550:

The rTg4510 mice are unlikely to be a model of Alzheimer's disease, since there are no mutant forms of tau in AD. Yes it is known today that not only familial AD, but also the spontaneous form of AD have a broad genetic background, but not mutations in the tau protein. Rather, this mouse line can be considered a model of frontotemporal dementia.

(9)  Suppl. Fig. S5:

What is the difference between B and C?
